# Hydrogen Sulfide Attenuates Neuroinflammation by Inhibiting the NLRP3/Caspase-1/GSDMD Pathway in Retina or Brain Neuron following Rat Ischemia/Reperfusion

**DOI:** 10.3390/brainsci12091245

**Published:** 2022-09-15

**Authors:** Kun-Li Yang, Wen-Hong Li, Ya-Jie Liu, Ying-Juan Wei, Yan-Kai Ren, Chen-Di Mai, Si-Yu Zhang, Yue Zuo, Zhen-Zhou Sun, Dong-Liang Li, Chih-Huang Yang

**Affiliations:** 1Department of Physiology, Sanquan College of Xinxiang Medical University, Xinxiang 453000, China; 2Department of Physiology, North Sichuan Medical College, Nanchong 637007, China; 3The Third Affiliated Hospital, Xinxiang Medical University, Xinxiang 453000, China; 4Department of Physiology and Pathophysiology, Xinxiang Medical University, Xinxiang 453000, China; 5Ineye Hospital, Chengdu University of Traditional Chinese Medicine, Chengdu 610000, China

**Keywords:** pyroptosis, neuron, GSDMD, NLRP3, casepase-1, retina, hydrogen sulfide, neuroinflammation, ischemia, reperfusion

## Abstract

Gasdermin D-executing pyroptosis mediated by NLRP3 inflammasomes has been recognized as a key pathogenesis during stroke. Hydrogen sulfide (H_2_S) could protect CNS against ischemia/reperfusion (I/R)-induced neuroinflammation, while the underlying mechanism remains unclear. The study applied the middle cerebral artery occlusion/reperfusion (MCAO/R) model to investigate how the brain and the retinal injuries were alleviated in sodium hydrogen sulfide (NaHS)-treated rats. The rats were assigned to four groups and received an intraperitoneal injection of 50 μmol/kg NaHS or NaCl 15 min after surgery. Neurological deficits were evaluated using the modified neurologic severity score. The quantification of pro-inflammatory cytokines, NLRP3, caspase-1, and GSDMD were determined by ELISA and Western blot. Cortical and retinal neurodegeneration and cell pyroptosis were determined by histopathologic examination. Results showed that NaHS rescued post-stroke neurological deficits and infarct progression, improved retina injury, and attenuated neuroinflammation in the brain cortexes and the retinae. NaHS administration inhibits inflammation by blocking the NLRP3/caspase-1/GSDMD pathway and further suppressing neuronal pyroptosis. This is supported by the fact that it reversed the high-level of NLRP3, caspase-1, and GSDMD following I/R. Our findings suggest that compounds with the ability to donate H_2_S could constitute a novel therapeutic strategy for ischemic stroke.

## 1. Introduction

Ischemic stroke has been the leading cause of mortality in more than 80% of Chinese provinces and has attracted mounting attention [1]. The restoration of blood supply followed by a period of ischemia that leads to injury, termed ischemia-reperfusion injury (IRI), is viewed as a characteristic of ischemic stroke [2]. Retina, as a developmental extension of the brain sharing various common pathophysiological alterations with the brain [3,4], is highly likely to be affected while cerebral stroke occurs. The loss of visual function due to apoptosis, necrosis, and pyroptosis has been extensively studied in several inflammatory disorders [5,6,7]. A study has also showed that visual dysfunction in stroke patients is frequently brought on by optic nerve damage and retinal thinning [8].

Cerebral I/R triggers neuroinflammation, which promotes the activation of immune cells and a series of inflammatory cascades in the brain [9] and the eye [10]. The inflammasome, a class of cytosolic protein complex, is essentially assembled before the neural-immune response initiation, and the NLRP3 inflammasome is the currently most thoroughly studied one. After inflammasome assembly, pro-caspase-1 is recruited and converted into the cleaved caspase-1, subsequently mediating the cleavage of gasdermin D (GSDMD) [11], which is a pyroptosis-executive protein whose activation can form pores on the plasma membrane and thereafter elicit cell lysis, i.e., pyroptosis [12,13,14].

NLRP3 is an impairing cellular function inflammasome, and the association between it and a number of neurodegeneration diseases has been extensively studied [15,16]. Amelioration of neurovascular damage and improved outcome in NLRP3-deficient mice of post-stroke, rendering NLRP3 essential toward a worse outcome following stroke [17]. Furthermore, the exhibition of neuroprotective effects against IRI in hispidulin by suppressing NLRP3-mediated pyroptosis has also been confirmed [18]. These findings indicate that NLRP3/caspase-1/GSDMD pathway contributes to stroke.

As a highly potent gasotransmitter, H_2_S is produced in the brain and other tissues and is involved in physiological processes including vasodilation, endocrine, and neuro-regulation [19,20,21]. Numerous studies have showed the value of H_2_S in inflammation inhibition in systematic diseases including valve calcification and acute renal injury [22,23]. It is also noteworthy that our earlier study has demonstrated that animals pretreated with a low concentration of H_2_S could attenuate IRI [24]. Additionally, retina-derived H_2_S shows a biologically protective capability [25,26], though the precise function of H_2_S in the retina remains unclear to date. Thus, the aim of next step is to determine the role of H_2_S in retinal inflammation as well as its underlying mechanisms. The aim of this study is to investigate whether NaHS, the supplier of H_2_S, can protect the cerebral cortex and the retina following I/R and whether it exerts neural protective ability by targeting the NLRP3/caspase-1/GSDMD pathway.

## 2. Materials and Methods

### 2.1. Animals

Beijing Vital River Laboratory Animal Technology (Beijing, China) provided Sprague–Dawley rats, male, weighed 250–320 g. All animals were housed at room temperature set at 23 ± 1 °C under a 12 h light/dark cycle and were allowed to move freely and received free food and water. The rats in this study include four groups (*n* = 25/group): the Sham group, the I/R group, the I/R + NaHS group, and the NaHS group. Both the I/R + NaHS group and the NaHS group received an intraperitoneal injection of 50 μmol/kg NaHS 15 min after ischemia or sham surgery. The same amount of NaCl was given to the Sham group and the I/R group. The Animal Care Committee of Sanquan College of Xinxiang Medical University approval this animal experiments (NO.SQLL-2019026).

### 2.2. The Middle Cerebral Artery Occlusion/Reperfusion (I/R) Model

Firstly, chloral hydrate was used to anesthetize rats (40 mg/100 g, i.p.). Secondly, the right external carotid artery, common carotid artery, and internal carotid artery were exposed. Thirdly, a silicon-coated monofilament with a diameter of 0.16 ± 0.02 mm was inserted through the external carotid artery and advanced into the internal carotid artery to achieve the middle cerebral artery occlusion (MCAO). Fourthly, upon 120 min of ischemia, the suture was removed to allow for a 24 h reperfusion. The rats of the Sham group and the NaHS group underwent identical surgical exposure procedures with the exception of the internal carotid artery occlusion.

### 2.3. Evaluation of Neurological Deficits and Behavioral Analysis

Twenty-four-hour application of the modified neurologic severity score (mNSS) was performed for the assessment of neurological deficits on the experimental rats following reperfusion. The mNSS includes reflex absence, motor tests, beam balance tests, and sensory tests and abnormal movements. The score of mNSS includes 0–18 points in which a larger score indicates greater neurological deficits.

### 2.4. TTC Staining

The rats were decapitated, and the brains were removed 24 h after I/R. The brains were frozen for 30 min and cut into 3 mm-thick sections. Following that, sections were fully stained using 2, 3, 5-triphenyltetrazolium chloride (TTC) (Sigma-Aldrich, Saint Louis, MO, USA) dissolved in 2% saline. The incubation conditions were 37 °C for 30 min. After triple washing with PBS, the sections were photographed immediately. Image J software (https://imagej.net/Fiji/Downloads, accessed on 30 May 2017, Bethesda, MD, USA) ultimately determined the infarct volume.

### 2.5. ELISA Assay for Inflammatory Cytokines

The right brain cortex and retina were removed from anesthetized rats, and protein extraction was performed after cleavage and homogenization. Protein quantification was performed with a bicinchoninic acid (BCA) protein assay kit (Beyotime Biotechnology, Shanghai, China). An ELISA assay kit was applied to measure the secretion levels of IL-1β and IL-18 (Rat IL-1β, Code:EK0393; Rat IL-18, Code:EK0592; Boster Biological Technology, Wuhan, China).

### 2.6. Western Blot Assay

The right brain cortex and retina protein samples (10 μg) were subjected to 8–12% SDS-PAGE and then transferred to PVDF membranes (Merk Millipore, Burlington, MA, USA). Blocked with 5% normal goat serum (NGS) for 2 h, the membranes were incubated with primary antibodies: rabbit Anti -NLRP3 (1:1000, Cat# ab263899; Abcam), rabbit anti-GSDMD (1:500, Cat# AF-4012; Affinity), rabbit anti-caspase-1 (1:1000, Cat# AF-4022; Affinity), and rabbit anti-β-actin (1:200; Cat# AF5003; Beyotime Biotechnology, Shanghai, China) at 4 °C for 24 h. The membranes were then incubated with anti-rabbit secondary antibodies (1:500; Cat# A0239; Beyotime Biotechnology, Shanghai, China) at 4 °C for 2 h. The protein signals were detected by Immobilon Western Chemiluminescent HRP substrate (Merk Millipore, Burlington, MA, USA) and analyzed using ImageJ analysis software (https://imagej.net/Fiji/Downloads, accessed on 30 May 2017, Bethesda, MD, USA). 

### 2.7. Brain Frozen Section and Retina Paraffin Section

Anesthetized rats were cardiac perfusion with 0.9% NaCl and 4% PFA. Rat brains and retinae were fixed in 4% PFA for 24 h. Rat brains were dehydrated in different concentrations of sucrose solutions (10%, 20%, and 30%). The brains were frozen and sliced into 40 µm-thick sections by freezing microtome (Leica CM1950, Leica, Wetzlar, Germany). Rat retinae were fixed in 4% PFA for 24 h, and then the retinae were dehydrated in 50%, 70%, 80%, 90%, and 100% alcohol solutions at 4 °C. The retinae were embedded in paraffin wax and sliced into 5 μm-thick sections by Ultra-Thin Semiautomatic Microtome (Leica 2245, Wetzlar, Germany).

### 2.8. Hematoxylin and Eosin Staining (HE)

After double washing with xylene solution, retina sections were then stained with HE (Cat# G1102, Solarbio, Beijing, China). The images were captured with a microscope (Laika, MD500, Wetzlar, Germany). ImageJ software collected the retinal thickness (https://imagej.net/Fiji/Downloads, accessed on 30 May 2017, Bethesda, MD, USA).

### 2.9. TUNEL Staining

The detection of retinal apoptotic cells was performed using a TUNEL Assay Kit (Cat# G1501, Wuhan servicebio technology, Wuhan, China). Briefly, retina sections were immersed in the TUNEL mixture for 2 h at room temperature, followed by staining with DAPI. Under the fluorescence microscopy, photographs were taken (A1+, Nikon, Japan; NIS Elements AR software, Nikon, Japan) that collected the positive signals (https://imagej.net/Fiji/Downloads, accessed on 30 May 2017, Bethesda, MD, USA).

### 2.10. Immunofluorescence

After three consecutive washes with PBS, brain sections were incubated in 0.5% triton X-100 for 30 min at room temperature. After double washing with xylene solution and being dehydrated in 50%, 70%, 80%, 90%, and 100% alcohol solutions, the retina sections were incubated in the sodium citrate solution at 95 °C for 15 min. Retina sections were incubated in the 0.5 % Triton X-100 at room temperature for 1 h after cooling the slide. The brain sections and retina sections were blocked with 5% NGS for 1 h and then were incubated with primary antibodies overnight: mouse anti-GSDMD (1:200, Cat# sc-393581, Santa Cruz Biotechnology, Dallas, TX USA), and rabbit anti-NeuN (1:1000, Cat# ab177487, Abcam, UK) at 4 °C, followed by incubation with appropriate Alexa Fluor-488-conjugated rabbit secondary antibodies (1:1000, Cat# S0018, Affinity, San Francisco, CA USA), Alexa Fluor-594-conjugated mouse secondary antibodies (1:1000, Cat# S0005, Affinity, San Francisco, CA USA), and DAPI (Cat# C1005, Beyotime Biotechnology, Beijing, China). Brain section immunofluorescence images were captured with a laser confocal microscope (A1+, Nikon, Japan) and Viewer software (NIS Elements AR, Nikon, Japan). After retina section immunohistochemical staining, under the fluorescence microscopy, photographs were taken (NI-U, Nikon, Japan;).ImageJ software collected the positive signals (https://imagej.net/Fiji/Downloads, accessed on 30 May 2017, Bethesda, MD, USA).

### 2.11. Statistical Analysis

All data were acquired from three independent experiments and are expressed as mean ± SD. Statistical analysis was performed using GraphPad Prism 8.0 (San Diego, CA, USA). An unpaired two-tailed Student’s t-test was used to compare data from two groups, and ANOVA was performed for multiple group comparison. Differences were considered significant when *p* < 0.05. 

## 3. Results

### 3.1. NaHS Rescued Post-Stroke Neurological Deficits and Inhibited Infarct Progression

Rat I/R model was conducted to investigate the function of NaHS against ischemia-reperfusion injury. The experimental procedures are presented in Figure 1a. Briefly, 15 min after MCAO, rats were given NaHS and the suture was removed 2 h later to allow for a 24 h reperfusion. The rat brain was collected to measure the infarct size. TTC staining suggested subcortical tissue of the I/R ipsilateral brain was partially infarcted, and NaHS administration significantly reduced infarct volume by 8.64% compared to I/R rats receiving NaCl (Figure 1b,c). The neurological impairment of the rat was evaluated using the mNSS. The I/R group had a higher neurological deficit score, which was decreased in the I/R + NaHS group, suggesting that NaHS could protect rats from ischemia-reperfusion injury (Figure 1d).

### 3.2. NaHS Improved Retinal Injury Induced by I/R

The protective capacity of NaHS on the retina was then assessed. The results showed that NaHS prevented interplexiform layer (IPL) attenuation and cell loss in ganglion cell layer (GCL). I/R-damaged retinae displayed a notable reduction in overall thickness. Compared with the Sham group, the overall thickness of the retinae decreased by 124.23 µm in the I/R group. The I/R + NaHS group considerably decreased this figure to 58.36 μm, attenuating the I/R-induced decline in retinal thickness (Figure 2a). Moreover, the TUNEL assay revealed that cell apoptosis was significantly promoted in the I/R group, while it was inhibited by NaHS in retinal GCL (Figure 2b).

### 3.3. NaHS Improved Neuroinflammation by Suppressing I/R-Induced Inflammasome-Dependent Pyroptosis in Rat Brain Cortex

Next, we examined the mechanisms underlying the suppressive effect of NaHS on neuroinflammation. Firstly, the production of proinflammatory cytokines among groups were compared. Cerebal I/R enhanced the expression of IL-1β and IL-18 in the rat brain cortex, which was reversed by NaHS (Figure 3c). A process that drives neuroinflammation in stroke, for example, a rise in caspase-1 cleavage and release of IL-1β, results from the activation of the NLRP3. Investigation on whether there was an association between the NLRP3 inflammasome pathway and NaHS-mediated attenuation of neuroinflammation was conducted. A significant increase in NLRP3, pro-caspase-1, and cleaved-caspase-1 was observed in the brain cortex tissues from I/R rat, suggesting the activation of the NLRP3 inflammasome pathway. However, the NLRP3 activation was suppressed in the I/R + NaHS group. Notably, I/R activated the downstream pyroptosis effector molecule GSDMD, and NaHS administration also lowered GSDMD expression (Figure 3a,b). The results above imply that NaHS could prevent I/R-induced inflammasome activation and pyroptosis.

### 3.4. NaHS Improved Neuroinflammation by Suppressing I/R-Induced Inflammasome Mediated Pyroptosis in Rat Retina

We conducted experiments to detect pyroptosis in the rat retina and obtained consistent results, further confirming the suppressive effect of NaHS on the canonical inflammasome pathway induced by cerebral I/R. We discovered that the NaHS + I/R group considerably decreased the rise in retinal production of IL-1β and IL-18 compared to the I/R group (Figure 4c). Further supporting the findings in the brain, treatment with NaHS reduced the NLRP3 expression level, inhibited the activation of caspase-1, and further declined GSDMD cleavage in the retina (Figure 4a,b).

### 3.5. NaHS Prevented Neuron Pyroptosis in Rat Brain Cortex after Cerebral I/R Injury

When compared with the sham-operated group, the number of Neu-N positive cells considerably dropped in the I/R group, whereas it significantly increased in the I/R + NaHS group, indicating that NaHS had the potential to increase neuron survival after I/R (Figure 5a,b). Immunofluorescence found that GSDMD-positive neurons were more prevalent in the Ischemic junction 24 h after MCAO, and the trend was notably reversed by NaHS administration (Figure 5a,c). This demonstrated that NaHS could inhibit the brain cell pyroptosis following I/R. The ratio of GSDMD and Neu-N double-positive cells to Neu-N positive cells was higher in the I/R group and decreased in the I/R + NaHS group (Figure 5a,d). All results suggested that NaHS could be able to prevent I/R-induced neuron pyroptosis.

### 3.6. NaHS Prevented Neuron Pyroptosis in Rat Retina after Cerebral I/R Injury

To further confirm the suppressive effect of NaHS on neuron pyroptosis after cerebral I/R injury, similar experiments were carried out using retina samples. Consistent with the results acquired above, we discovered that I/R dramatically decreased the number of Neu-N positive cells in the retina, whereas NaHS greatly increased the number of Neu-N positive cells in the retinal GCL region (Figure 6a,b). These findings showed that NaHS could improve the survival rate of retinal neurons after I/R. Moreover, 24 h after MCAO, GSDMD-positive neurons were more prevalent in the I/R-damaged retina, and NaHS reversed this trend (Figure 6a,c). The ratio of GSDMD and Neu-N double-positive cells to Neu-N positive cells was much more than that in the I/R group and decreased in the I/R + NaHS group (Figure 6a,d). These results suggested that NaHS could also inhibit I/R-induced neuron pyroptosis in the retina.

## 4. Discussion

H_2_S or its supplier NaHS has protective effects against inflammation in the murine cardiac arrest model and rat stroke model [24,27]. In this manuscript, we further explored the specific locations where H_2_S exerts neuroprotective effects in cerebral I/R injury and related mechanisms. The results suggest that H_2_S not only alleviates neuroinflammation in the cerebral cortex after I/R injury, but also ameliorates retinal neuroinflammatory injury, and both mechanisms are closely related to NLRP3/caspase-1/GSDMD pathway-mediated neuronal pyroptosis.

The anti-inflammatory effect of H_2_S treatment on brain I/R injury has been confirmed [28]; however, the exact mechanism is still unknown. To fill this research gap, the following experiment was conducted. First, this research verified the therapeutic effect of H_2_S on cerebral I/R, and several experimental results including TTC staining and neurological scoring confirmed the significant inhibition of the I/R injury process by H_2_S. Consistent with previous studies, the observed therapeutic effect of H_2_S on cerebral I/R was mainly focused on the inhibition of neuroinflammation [29]. In contrast, not only was the inhibitory effect reflected in the cortex of rats, but it also showed such protection in the retina. Our results suggest that H_2_S downregulates the expression of IL-1β and IL-18 in cortical and retinal tissues. Recruitment and activation of inflammatory cells and increased secretion of pro-inflammatory cytokines in the inflammatory response are thought to be the principal causes of I/R-induced pathological changes in the cortex; however, the underlying pathophysiological mechanisms involved are more complex. Among them, inflammasomes, a newly discovered cytoplasmic protein complex in recent years, have been found to be actively involved in the neuroinflammatory response in I/R [30]. It has been demonstrated that inflammasomes can trigger neuronal death through multiple mechanisms, which in turn exacerbates cerebral I/R injury [31]. We hypothesized that the anti-inflammatory effect of H_2_S is closely related to inflammatory vesicles and verified it in the present study. NLRP3 is the most comprehensively studied and representative inflammasome. Caspase-1 and GSDMD are the key downstream effector molecules upon NLRP3 activation. Our results showed that the expression of NLRP3 and its downstream molecules—caspase-1/GSDMD—were significantly reduced in the cortical sites of I/R rats compared to I/R rats after administration of NaHS. The trend of NLRP3/caspase-1/GSDMD expression in the retina was also consistent. This result implies a significant modulatory effect of H_2_S treatment on inflammasomes and provides the first evidence that H_2_S ameliorates the NLRP3 inflammasomes-related mechanism of retinal damage in I/R rats.

Second, irreversible neuronal damage after brain I/R injury is one of the important inducements of permanent brain function deficits. Cellular necrosis and apoptosis as the main mechanisms of neuronal damage after cerebral I/R, however, discovery of NLRP3 inflammasomes promoted a third mechanism of cell death, pyroptosis, in addition to necrosis and apoptosis [32]. Subsequently, it was demonstrated in the literature that NLRP3 inflammasomes can exacerbate neural injury by regulating neuron pyroptosis after brain I/R injury [33]. However, whether the regulation of NLRP3/caspase-1/GSDMD pathway by H_2_S also affects neuronal pyroptosis after brain I/R has not been reported. The experimental results of this manuscript showed that NaHS treatment significantly reduced the pyroptosis of cortical neurons after I/R injury in rat brain. In particular, significant inhibition of pyroptosis in rat retinal neurons induced by I/R injury was also observed. The reduced level of cortical and retinal neuronal pyroptosis implies the suppression of inflammatory response. This further explains why NaHS treatment was capable of significantly improving brain I/R damage and inhibiting the progress of cerebral infarction in rats. In conclusion, this study provides the first experimental basis for NaHS to inhibit NLRP3/caspase-1/GSDMD inflammatory vesicle pathway activation, alleviate cortical and retinal neuronal pyroptosis, and improve brain I/R.

Although this study initially demonstrated the therapeutic effect of NaHS treatment on cortical and retinal neuroinflammation after cerebral I/R and the modulatory effect on neuronal pyroptosis, the exploration of the mechanism is still lacking in depth. Briefly, future studies need to further inhibit the expression of the NLRP3/caspase-1/GSDMD pathway and then examine the modulatory effects of NaHS treatment on neuronal necrosis and neuroinflammation after I/Rin rats. This can provide a more convincing account of the rationale and mechanism of NaHS treatment of cerebral I/R. Overall, this research firstly expands the understanding of the anti-inflammatory effects of NaHS. It argues for the first time that the modulatory effects of NaHS on cortical and retinal neuronal pyroptosis and inflammatory responses after I/R injury are associated with the NLRP3/caspase-1/GSDMD pathway. Second, the present study also provides a solid theoretical basis in further promoting the application of NaHS in the treatment of clinical brain I/R injury.

## 5. Conclusions

To conclude, the present study demonstrated that NaHS protects against I/R-induced cortical and retinal injury, which is related to the downregulation of the NLRP3/caspase-1/GSDMD pathway. Our research showed that compounds with the ability to donate H_2_S can be considered a potential therapeutic strategy for the treatment of cerebral ischemia reperfusion.

## Figures and Tables

**Figure 1 brainsci-12-01245-f001:**
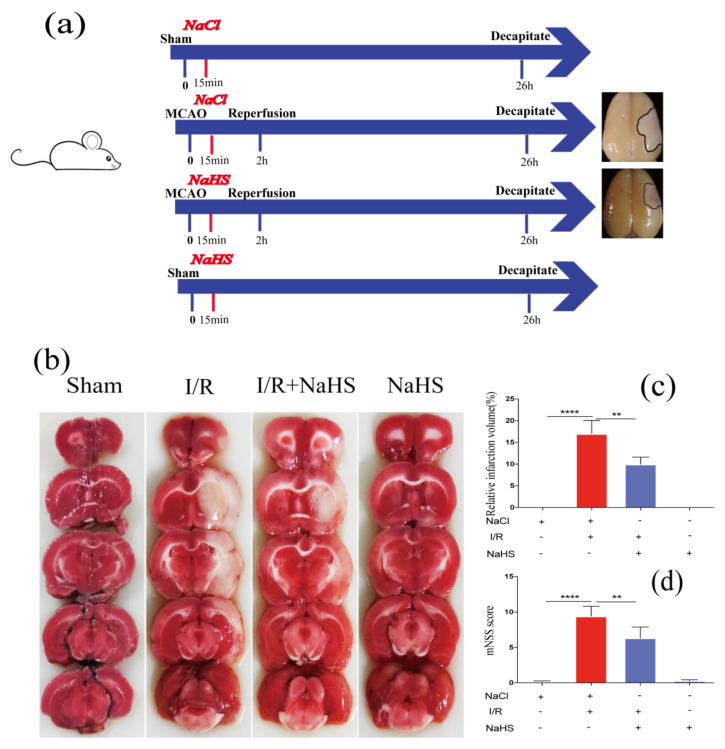
NaHS rescued neurological deficits and ischemic infarction of I/R rat. (**a**) The schematic of experimental procedures. (**b**) Images of brain TTC staining in each group. (**c**) The quantitative analysis of infarct volume. (**d**) Neurological deficit was evaluated by the mNSS scoring system 24 h post I/R. Data are presented as the means ± SD, *n* = 5, ** *p* < 0.01, **** *p* < 0.0001.

**Figure 2 brainsci-12-01245-f002:**
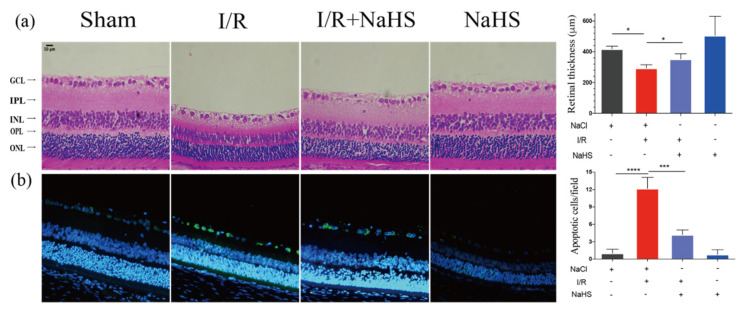
NaHS alleviated I/R-induced retinal neurodegeneration. (**a**) Images of hematoxylin and eosin staining and overall thickness of retinae quantitative analysis. NaHS improved the retinal thinning induced by cerebral I/R. (**b**) Representative images of TUNEL staining and quantitative analysis of TUNEL+ (apoptotic, green) cells. NaHS decreased apoptotic cells in GCL. Scale bar = 50 μm. Data are presented as the means ± SD, *n* = 5, * *p* < 0.05, *** *p*< 0.001, **** *p*< 0.0001. Ganglion cell layer—GCL; interplexiform layer—IPL; Inner Nuclear Layer—INL; outer plexiform layer—OPL; outer nuclear layer—ONL.

**Figure 3 brainsci-12-01245-f003:**
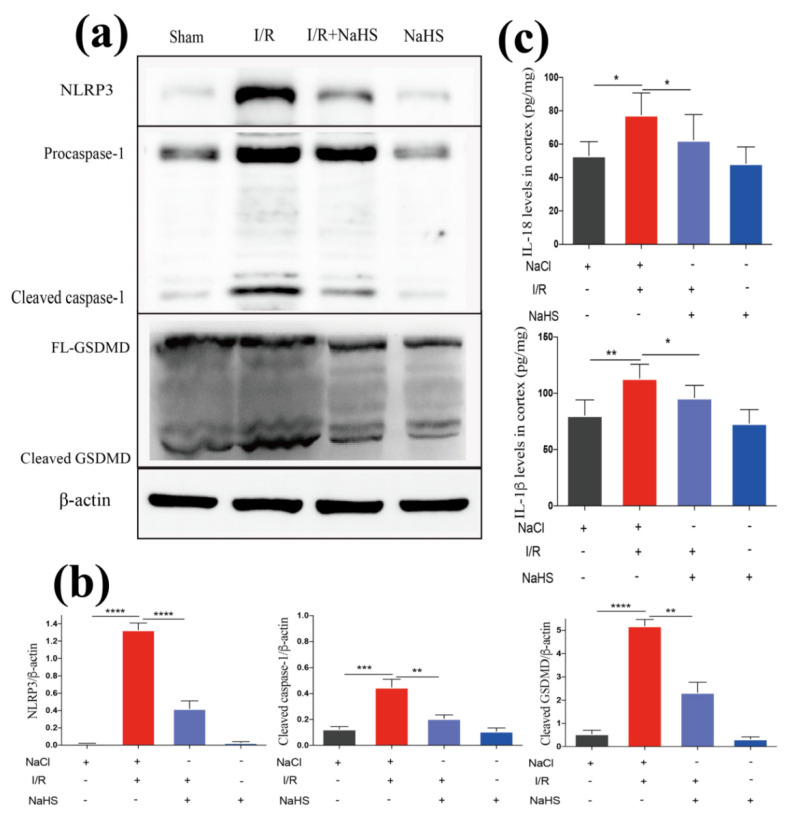
NaHS suppressed inflammasome-dependent pyroptosis in the rat brain cortex after I/R. (**c**) ELISA analysis of proinflammatory cytokines in the brain cortex from I/R rat (*n* = 5). (**a**) Western blot analysis of NLRP3, caspase-1, and GSDMD in the brain cortex. (**b**) Quantitative analysis of expression levels of NLRP3, caspase-1, and GSDMD in the brain cortex (*n* = 3). Data are represented as means ± SD, * *p* < 0.05, ** *p* < 0.01, *** *p* < 0.001, **** *p* < 0.0001.

**Figure 4 brainsci-12-01245-f004:**
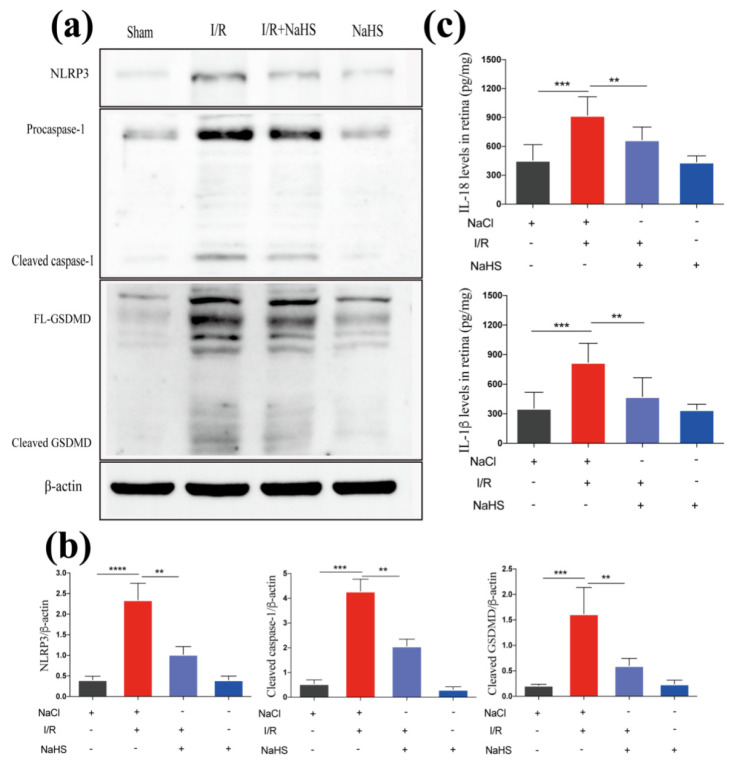
NaHS suppressed I/R-induced inflammasome-dependent pyroptosis in the rat retina. (**c**) ELISA analysis of proinflammatory cytokines in the retina from I/R rat (*n* = 5). (**a**) Western blot analysis of NLRP3, caspase-1, and GSDMD in the retina. (**b**) Quantitative analysis of expression levels of NLRP3, caspase-1, and GSDMD in the retina (*n* = 3). Data are represented as means ± SD, ** *p* < 0.01, *** *p* < 0.001, **** *p* < 0.0001.

**Figure 5 brainsci-12-01245-f005:**
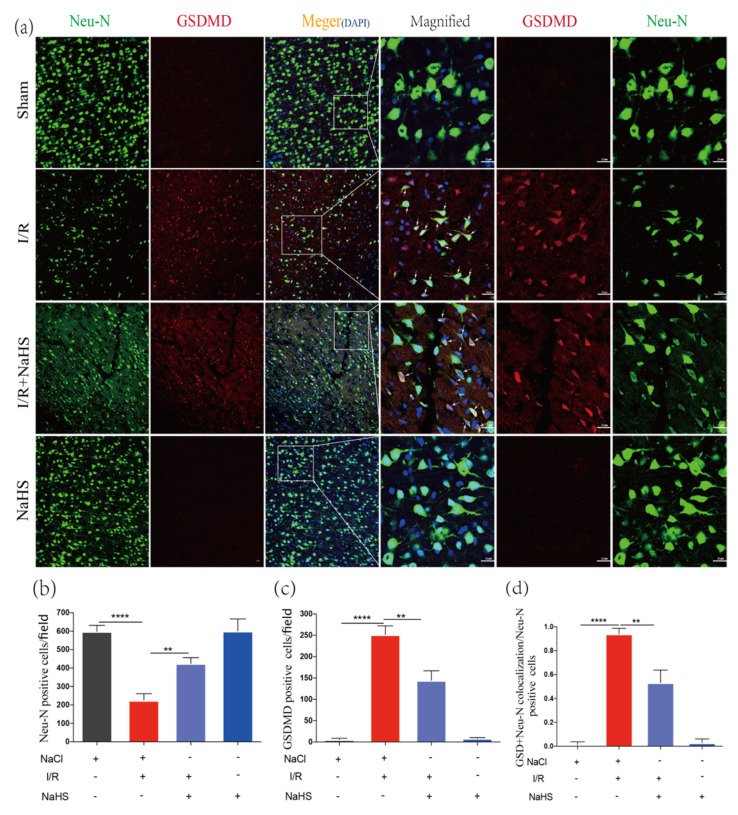
NaHS inhibited I/R-induced neuron pyroptosis in the rat brain cortex. (**a**) Immunofluorescence staining of Neu-N and GSDMD in the peri-infarct region. Scale bar = 50 μm. Scale bar = 25 μm in magnified images. Arrows indicate Neu-N and GSDMD double-positive cells. (**b**) Analysis of Neu-N Positive cells in the brain cortex (*n* = 3). (**c**) Analysis of GSDMD-positive cells in the brain cortex (*n* = 3). (**d**) Analysis of the ratio of the Neu-N and GSDMD double-positive cells to the Neu-N positive cells in the brain cortex (*n* = 4). Data are represented as means ± SD. ** *p* < 0.01, **** *p* < 0.0001.

**Figure 6 brainsci-12-01245-f006:**
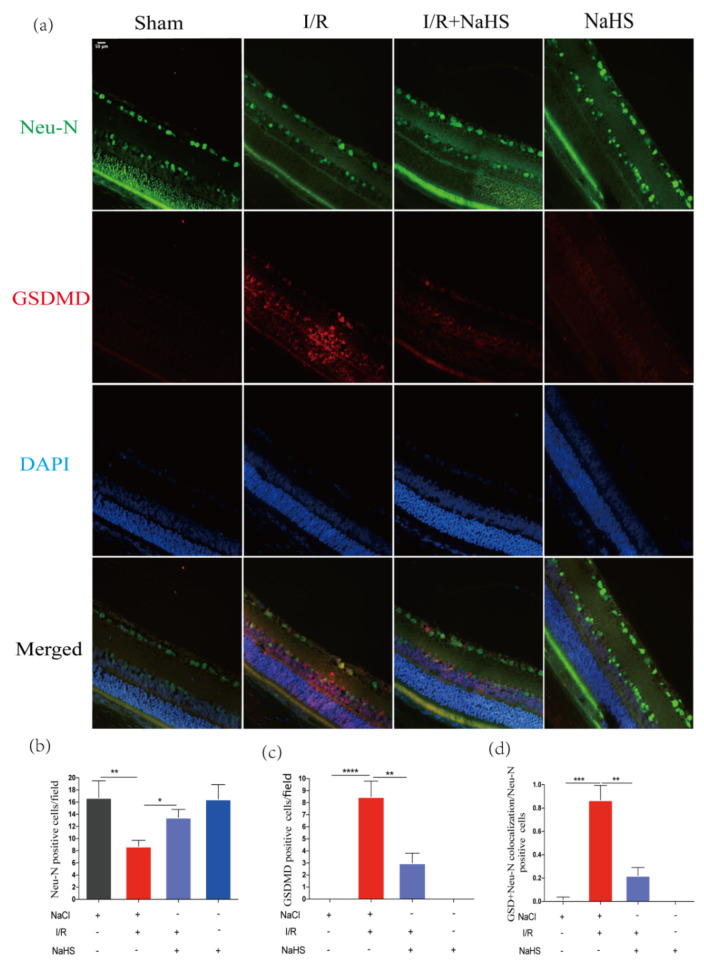
NaHS inhibited neuron pyroptosis in the rat retina after I/R. (**a**) Immunofluorescence staining of Neu-N and GSDMD of in retinal GCL region. Scale bar = 50 μm. (**b**) Analysis of Neu-N Positive cells in the retinal GCL region (*n* = 3). (**c**) Analysis of GSDMD-positive cells in the retina GCL region (*n* = 3). (**d**) Analysis of the ratio of the Neu-N and GSDMD double-positive cells to the Neu-N^+^ cells in the retina GCL region (*n* = 4). Data are represented as means ± SD. * *p* < 0.05, ** *p* < 0.01, *** *p* < 0.001, **** *p* < 0.0001.

## Data Availability

The datasets generated and/or analyzed during the current study are not publicly available due we have unpublished studies from this data but are available from the corresponding author on reasonable request.

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
