# Peer review of "Hydrogen Sulfide Attenuates Neuroinflammation by Inhibiting the NLRP3/Caspase-1/GSDMD Pathway in Retina or Brain Neuron following Rat Ischemia/Reperfusion"

_brainsci, 2022, doi:10.3390/brainsci12091245_

Round 1
Reviewer 1 Report
In the manuscript brainsci-1888452 authors showed that hydrogen sulphide attenuates Ischemia/reperfusion by modulating NLRP3/caspase-2/GSDMD pathway mediated neuroinflammation in rats. The results are interesting and the figures are very clear, however, there are some questions that should be clarified in the text. Here are my suggestions for improving the manuscript.
Major concerns
1. There is a scope to improve the paragraph 3 of the introduction. The first sentence of the paragraph is deviating from the main aim of the study; the emphasis should be given on involvement of NLRP3-mediated proptosis on IRI (if evidence available) or other neurological disorders. Few sentence additionally (before sentence one) might be enough.
2. Panel (a) of the figure 1 seems incomplete without Sham and NaSH groups, hence same should be added to complete the schematic experimental procedure.
3. The procedure for the sample preparation for ELISA and western is same, hence detailed procedure should only be given under “ELISA assay” heading and should be mentioned as “given above for ELISA assay” under “Western blot assay”. The brand name is of ELISA kit not of IL-1β and IL-18, thus the same should be given just after ELISA assay kit.
4. Why only two cytokines (IL-1β and IL-18) have been selected in the current study, while you could have selected more inflammatory markers e.g., cytokine (IL-6, IL-12) and TNF-α?
5. The procedures for tissue preparation for H&E, TUNEL and immunofluorescence should be written precisely, there is no need to repeat every time. Details of the microtome (used for sectioning) should also be given. The electronic link of Image J software should be provided instead in NIH.
6. The discuss need to be rewritten, your results should be explained in more detail with the help of existing research in the field.
7. Authors find a way to minimize the duplication as the manuscript is showing about 35% text similarity on Turnitin.
Minor comments
1. In introduction (line 52) “GSDMD” after citation [11] should be replaced with “which”. “shows” should be replaced with “showed” (line 65). Authors need to check “neural protective” or neuroprotective, which one will apply?
2. The ethical clearance code should be given in under the “Animals” subheading.
Overall comments:
The author has done a good work with appropriate material and methods, and results. The discussion section needs further improvement.
Author Response
Dear editor:
Thank you for your questiones .I will answer your question one by one .
Answer to Major concerns
1: The paragraph 3 of the introduction is rewritten.You can review it in the uploaded manuscript.
2: Sham and NaSH groups have been added to complete the schematic experimental procedure.
3: The brand name of ELISA assay kit have been given.The procedure for the sample preparation for western have been deleted.
4:According to the literature report,the main inflammatory factor released by the pyroptosis pathway is IL-1β and IL-18, so we only detected IL-1β and IL-18.It is to prove that hydrogen sulfide can improve brain injury after ischemia by inhibiting pyrotosis.
5:“2.8 Retina paraffin section”and “2.9.Hematoxylin and Eosin Staining” have been added in the Materials and Methods part .The slicing method has been described in detail.
6: The discuss have been rewritten,you can review it in the uploaded manuscript.
7: The revision manuscript is showing about 27% text similarity on Turnitin no w.
Answer to Minor comments
1:The words have been redress in introduction .I think this word (neuroprotective) is better .
2: Animal experiments were performed in accordance with the Animal Care Committee Sanquan Collage of Xinxiang Medical University(NO.SQLL-2019026).

Reviewer 2 Report
The article by Yang et al. discloses an important issue regarding the influence of H2S donors on the regulation of the NLRP3 / caspase-1/GSDMD pathway.
Research will contribute to the development of the research area related to H2S donors.
Minor:
1) Conclusions, lines 329-330. ‘Our research showed that H2S may potentially be a novel therapeutic agent for the treatment of stroke’.
It is highly unlikely that H2S alone would be approved by any drug regulatory agency such as the FDA, NMPA or EMA, due to its high toxicity. For that reason, compounds with the ability to provide H2S, such as sodium polysulthionate or sodium thiosulfate, are currently in clinical trials in human (eg. NCT02899364). Therefore, I suggest that the authors change the sentence provided in the conclusions into eg, 'Our research showed that compounds with the ability to donate H2S can be considered as a potential therapeutic strategy for the treatment of stroke’.
2) The same refers to a sentence provided in the abstract: line 30: ‘Our findings suggest that NaHS might be a novel therapeutic agent for stroke’.
So far, NaHS has not been approved by the regulatory agencies for various reasons. Therefore, I suggest rewriting the sentence “Our findings suggest that compounds with the ability to donate H2S could constitute a novel therapeutic strategy for ischemic stroke”.
3) Figure 1A. Please improve the quality of a scheme by deleting the frame around mouse icon
Author Response
Dear editor:
Thank you for your ideas .The manuscript have been modified according to your suggestion.Figure 1A also have been modified .Please see the attachment
